# Effectiveness of the “Mente Sana [Healthy Mind]” Cognitive Training Program for Older Illiterate Adults with Mild Cognitive Impairment

**DOI:** 10.3390/geriatrics5020034

**Published:** 2020-05-23

**Authors:** Yaneth del Rosario Palo Villegas, Andrea Elena Pomareda Vera, María Elena Rojas Zegarra, M. Dolores Calero

**Affiliations:** 1Professional School of Psychology, Universidad Nacional de San Agustín de Arequipa, Calle Santa Catalina 117, CP: 04000 Arequipa, Peru; mrojasze@unsa.edu.pe; 2Mind, Brain, and Behavior Research Center, University of Granada, Avenida del Hospicio, CP: 18071 Granada, Spain; mcalero@ugr.es

**Keywords:** older adults, illiterate, cognitive decline, cognitive training

## Abstract

Aging can lead to functional and cognitive alterations, sometimes limiting older adults in their social development, especially illiterate groups of older adults who receive poor attention from healthcare systems. In this context, the present investigation proposes the cognitive training program “MENTE SANA [HEALTHY MIND]” to improve the cognitive functions of illiterate older adults in Arequipa (Peru). It is a type of quasi-experimental research with a pre-test/post-test design with a homogenous control group. The sample was made up of adults 60 years old and above and of female gender. The Montreal Cognitive Assessment (MoCA) test was used to detect the level of cognitive decline in illiterate older adults. The 50-sessions program was applied to all the older adults with mild cognitive impairment that were selected for the study, on a daily basis. It was found that the tested group improved their cognitive functions compared to the control group. These results help to propose adapted cognitive training programs for illiterate people.

## 1. Introduction

It is estimated that the cases of dementia will increase in developing countries such as Peru, from 7.4% to 12% in 2025, and although these data could vary [1], they suggests a higher incidence of neurodegenerative diseases. Therefore, it is essential to prevent the cognitive deterioration of older adults progressing rapidly, particularly those people who have not received a formal education, as is the case of illiterate older adults. This is especially true for people with mild cognitive impairment, because they have a greater probability of responding to training programs as their cognitive functions such as memory, attention, etc., are still preserved. It is important to detect this disorder in our population to prevent serious cognitive decline such as Alzheimer’s disease. In Arequipa, an inversely proportional relationship was found between cognitive decline and age, with a greater increase in impairment observed in the illiterate population [2].

Over the years and with the intention of providing a solution to the problems described above, there has been growing interest in working on the cognitive skills of older adults in order to intervene in their mental health. This has inspired numerous research studies on using cognitive and memory training programs with a meta-analysis of seventeen cognitive intervention clinical trials for Mild Cognitive Impairment (MCI). The results demonstrated that after the training, MCI patients improved significantly in both general cognitive thinking and self-assessments [3]. These research studies reflected the positive impact of the different cognitive training programs based on the premise that cognitive processes in old age are characterized by their neuronal plasticity, which has been found to be activated during old age by a reserve capacity [4]. In turn, the importance of this plasticity in older adults and its relationship with longevity and successful aging was demonstrated [5]. A study to analyze the presence of neuronal plasticity in older people who seemed to present cognitive deterioration, (all alterations of the superior mental capacities: memory, judgment, abstract reasoning, concentration, attention and praxis [6]), involving a series of limitations regarding the autonomy and quality of life of older adults, showed that despite this alteration, the program significantly improved cognitive ability, while plasticity was shown to be an important modulating variable in the improvement achieved [7]. Additionally, a meta-analysis of all randomized controlled trials (RCTs) published between 1970 and 2007 was also performed. Overall, 24 studies in which the effects of memory training in healthy older adults with mild cognitive impairment were identified and included in the analysis, demonstrating significant effects for immediate memory recall in older adults with mild cognitive impairment [8].

### Background

Various studies revealed that some factors are more likely to protect against the decline of cognition in old age; for example, the level of education. This assumes that illiterate people are vulnerable populations that generally perform worse than the literate in a variety of cognitive and neuropsychological tasks. These research studies revealed a relationship between low educational level and mild cognitive impairment, such as the one carried out with the aim of estimating the prevalence of mental disability in the elderly of the Playa-Cuba municipality and concluding that the increase in cognitive disability decreases as schooling increases [9]. Likewise, another study seeking to determine the functionality and degree of dependency in an educated Colombian older adult group and using this information for the design of a specific intervention program in which the majority received basic elementary level education, found that 41.7% presented severe cognitive impairment [10]. Similarly, other researchers, with the objectives of evaluating cognitive function in older people in China, identifying the relationship between cognitive function and different characteristics, and evaluating the efficacy of the intervention after six months of cognitive training, concluded that old age is associated with cognitive decline and that one of the factors most likely to protect against cognitive decline is a higher education level. Furthermore, cognitive training intervention is effective and may help decrease the deterioration of cognitive function in patients with MCI, and the link between intervention time and cognitive training significantly improves cognitive function [11]. Likewise, different functional imaging studies are showing that literacy and education influence the pathways used by the brain for problem-solving. The possible existence of partially specific neural networks as a consequence of greater levels of literacy supports the hypothesis that education affects not only the individual’s daily strategies but also brain networks. In a similar way, several studies using positron emission tomography and statistical parametric mapping or functional magnetic resonance imaging found additional differences between the literate and the illiterate in areas of the brain activated during language-based tests, i.e., the Perisilvian area of the left hemisphere of the brain [12].

For this reason, it is essential to examine the results of cognitive training in these illiterate populations made up of at least 750 million among young people and adults worldwide [13]. Few studies have been carried out on this topic. In one of the first studies, they carried out an investigation with the objective of comparing the effectiveness of a self-training program in inductive reasoning for older people with low levels of education, considering two different trainings; one guided by a tutor, and self-training, concluding that similarly positive effects were achieved in both scenarios [14]. Moreover, after applying a 12-week integrated program of cognitive training and physical activity for South Korean older adults with low education and mild dementia, they demonstrated that the use of these strategies increases cognitive capacity in the treatment group [15].

This means that it is possible to prevent and rehabilitate cognitive deterioration with training consisting of a set of actions aimed at maintaining or improving cognitive functioning through certain exercises. The purpose of this training is to work on those areas that are affected or that have been altered by disease or old age and on those people who, even without being affected, can benefit from these activities and can improve their functioning. The aim is slowing down, or, at best, reversing, the deterioration process, avoiding disconnection from their environment, increasing self-esteem and personal autonomy, avoiding stress, and improving the quality of life of the older adult and his or her family.

This research is directed to the illiterate older adult population, who due to personal and social reasons or the lack of opportunities, are very likely to have brains and cognitive systems that function atypically [12].

The study was carried out in two centers for the elderly where the development and application of a cognitive training program for illiterate older adults with mild cognitive impairment were proposed and adapted to the characteristics of their condition. That is why the general objective of this research was to determine the effectiveness of a cognitive training program for illiterate older adults with mild cognitive impairment. The proposed hypothesis was: The cognitive training program “MENTE SANA [HEALTHY MIND]” is effective for the rehabilitation or the delay of cognitive deterioration in illiterate older adults.

The specific objectives were: (1) Identify the level of cognitive decline in illiterate older adults (2) Apply a cognitive training program in illiterate older adults with mild cognitive impairment (3) Measure the level of cognitive decline in illiterate older adults with mild cognitive impairment after the application of the cognitive training program (4) Compare the results obtained between the control group and treatment group.

## 2. Materials and Methods

### 2.1. Participants

The participants of the study belong to a Peruvian Social Security Health Institution known as EsSalud, which is committed to the integral health of those insured. The III Elderly Center La Victoria and the II Elderly Center Miguel Grau were selected as generational meeting spaces aimed at improving the quality of life of older adults through the development of family integration, inter-generational, socio-cultural, recreational and productive programs with lifestyles for active aging. The people who attended these centers are retired older adult residents at home who regularly attend workshops including dance, aerobics, therapeutic gymnastics, TAI-CHI, computing, crafts, pastry, among others.

The sample consisted of 30 illiterate older adult women from 65 to 75 years of age originating from urban areas of medium socioeconomic status and selected non-probabilistically for convenience. The inclusion criteria considered were: (1) daily attendance so that they can persevere in the application of the cognitive training program, (2) agreement to participate voluntarily and comply with informed consent, (3) an initial score of mild cognitive impairment (17 to 21 points) in the Montreal Cognitive Assessment test, and (4) absence of psychiatric history, sensory disturbances, or chronic diseases.

### 2.2. Instruments

#### 2.2.1. Cognitive Impairment

In order to measure the level of cognitive impairment, the Montreal Cognitive Assessment (MoCA) [16] was used.

Its use was based on its simplicity and speed of application, in addition to its recognized reliability, dependability, validity, and adaptation in Latin America. The results of a study indicate that the MoCA test has a high inter-rater reliability of 0.91; good internal consistency (Cronbach’s Alpha = 0.85) and is able to discriminate different levels of cognitive impairment by showing adequate sensitivity (0.84) and specificity (0.71). This allows it to be postulated as useful and appropriate for screening cognitive changes in the Latin American population. Likewise, this analysis confirms that different values were found for the cut-off point of sensitivity and specificity according to the degree of education, so in subjects with primary or lower education, the score of 21/22 is considered as the cut-off point between normality and mild cognitive impairment [17,18].

It consists of 19 items and 8 cognitive domains with a maximum total score of 30 points that rates activities such as: visuospatial/executive (5 points), identification (3 points), memory (no points), attention (6 points), language (3 points), abstraction (2 points), delayed recall (5 points) and orientation (6 points).

#### 2.2.2. “MENTE SANA [HEALTHY MIND]” Cognitive Training Manual

In order to carry out the application of the cognitive training program, a manual for illiterate older adults was used, designed by the authors of this article and validated by the judgment of three expert clinical psychologists with experience in older adults who passed the MENTE SANA cognitive training program through an evaluation matrix in which various criteria were rated. These include the precision of the instructions, the size, and sharpness of the images, the distribution of space, the dynamics of the sessions, the preventive purpose of the manual, the relevance and necessity of the sessions, the intervention at the different levels of the individual (affective, social and cognitive), the specificity in the design of the manual for illiterate older adults, the duration of each session, the progressive difficulty of the sessions, the similarity of the exercises with the cultural reality of the participants and their suitability for them to comply with training cognitive skills (visuospatial/executive, identification, memory, attention, language, abstraction, delayed memory and orientation).

The MENTE SANA cognitive training manual has 50 sessions with a duration of approximately 25 min each, in which numerous exercises were performed, such as process sequence, copying and drawing images, identifying objects of everyday life, retaining information in the short term, copying graphic patterns, expressing feelings and emotions through words, searching for patterns, interpreting sentences, long-term retention of information, determining the position and movements of the body in space and time, solving mazes, narrating and creating stories, telling anecdotes, identifying differences and similarities, teaching antonyms and synonyms, identifying places, learning holiday dates, identifying people, professions, animals, naming and describing objects and doing manual work. The sessions were led by two experts who instructed the participants in the development of each session of the manual. The implementation of the cognitive training program lasted 2 and a half months and was carried out in small groups of approximately 3 to 5 people.

#### 2.2.3. Procedures

This was implemented based on accessing the institution’s database to identify the illiterate older adults. The distribution of the groups took into account that participants residing in different centers for the elderly could not exchange information about the exercises trained during the sessions. Thus it was determined that the control group was made up of those insured by the II Miguel Grau Senior Center and the treatment group by those insured by the III La Victoria Senior Center where the independent variable (the “MENTE SANA” cognitive training program) was manipulated in order to check its effect on the dependent variable (mild cognitive impairment), taking place in a natural context [19].

Firstly, after obtaining the research approval by the EsSalud Ethics Committee and the Research Committee of the Universidad Nacional de San Agustín de Arequipa and EsSalud, the signatures of the informed consents were collected and the participants were divided in two groups: 15 older adults from III La Victoria Senior Center were assigned to the treatment group (GT) that received “MENTE SANA cognitive training”, and 15 others to the control group (GC) belonging to the II Miguel Grau Senior Center. Both groups were submitted to the pretest evaluation in order to determine the level of cognitive impairment. Subsequently, after the application of 50 daily sessions of approximately 25 min each during a two-and-a-half-month period, a second evaluation was implemented to the total sample to check whether cognitive training had any rehabilitative effect on the treatment group participants compared to the control group.

### 2.3. Method and Data Analysis

A quasi-experimental design of two intact non-randomized equivalent groups was implemented (control group and treatment group) because the participants were assigned to each group according to the center for the elderly they attended. Figure 1 shows the consort flow diagram with which the longitudinal design phases are specified.

The data were analyzed using the Statistical Package for Social Sciences (SPSS, 2008), version 23. Based on the analysis of normality of the sample (Kolmorogov–Smirnov test and homogeneity analysis of variances of Levene) and its corresponding normal distribution, a general linear model of repeated measures was obtained (pre and post) and compared between groups (treatment and control).

The following analyzes were carried out in detail: (1) Classification of the level of cognitive decline in illiterate older adults of the III La Victoria Senior Center and II Miguel Grau Senior Center. EsSALUD. (2) Classification of the cognitive deterioration level in illiterate older adults of the III La Victoria Senior Center and II Miguel Grau-ESSALUD Senior Center after applying the cognitive training program. (3) A general linear model of repeated measures (pre and post) was obtained and a comparison between groups (treatment and control) was performed. (4) Student’s *t*-test was applied between pre- and post-tests for each of the cognitive domains evaluated by the MoCA.

## 3. Results

Of the distribution percentage of the normality criteria that were obtained when applying the MoCA method to illiterate older adults with mild cognitive impairment in the III La Victoria-Essalud Senior Center, in the pretest stage, 100% (15) of the control group was identified with mild cognitive impairment, and for the treatment group, 93.3% (14) was identified with mild cognitive impairment. Likewise, in the evaluation of mean values, it was observed that in the pre-test stage, the control group obtained a mean value of 18.60 points (and a standard deviation of 1.24 points) less than the treatment group with a mean value of 19.13 and a standard deviation of 1.35. These differences were identified in both control and treatment groups in the pre-test and, according to the statistical Student’s *t*-test, these were not considered significant with a value *p* = 0.271. Therefore, the initial stage of the experimental study started with common results in which both groups did not have significant differences in relation to the MOCA evaluation and most of them presented mild cognitive impairment.

Consequently, in Table 1 it can be observed that the intra-subject effects indicate that there are significant differences between the pre- and post-test between the two groups. The intra-subject pre-post differences (F = 14.641; *p* = 0.0001) reflect significant changes in the two groups between the two evaluation times (pre and post); changes that are different for each group (F = 45.08; *p* = 0.0001). These also appear as inter-group differences (F = 20.52; *p* = 0.0001) with a very high effect size (ETA = 0.989) and a power of 1.000.

This demonstrates that there is no significant improvement in cognitive functions in the control group participants, while in the treatment group there is evidence of significant improvement in the cognitive functions that existed in the participants after having implemented the cognitive training program, which led to an increase of 6.5 points on average in the test used. As can be seen, the impact sizes are moderate to considerable, and the power is very high for both intra-subject effects (change in the treatment group from pre- to post-tests) and for inter-subjects effects (differences between groups after treatment application). Student’s *t*-tests between the domains evaluated by the MoCA (Table 2) show significantly positive results between pre- and post-treatment for the visuo-spatial/executive treatment group (t = 4.56, *p* = 0.001) identification (t = 3.15, *p* = 0.004), attention (t = 3.94, *p* = 0.001), language (t = 8.20, *p* = 0.0001), and orientation (t = 2.53, *p* = 0.018), with the differences for the remaining scales being insignificant.

Figure 2 shows the differences between the mean scores in each cognitive domain evaluated by the MoCA in the control and treatment groups in the post-test, reflecting in their mean values by the evaluated area that cognitive training had an impact, since the scores in the treatment group are significantly higher than in the control group.

## 4. Discussion

The MENTE SANA [HEALTHY MIND] program applied to illiterate elderly people with mild cognitive impairment achieved a very significant improvement in their cognitive performance, which in average values represents 6.5 points, reflecting in the treatment group that initially, according to the MOCA method, 100% (14) of the participants were identified with mild cognitive impairment and after training 46.7% (7) were identified with mild cognitive impairment, and 53.3% (8) identified as normal. That is, 53.3% of the old adults improved their initial cognitive status to normal. These subjects significantly increased their cognitive execution at a level equal to or greater than other MCI elderly people with an educational level [3,20] higher than that expected given their initial level. [21]. When analyzing the effects between groups in the different dominion of the MoCA between the treatment and control groups, it can be seen that there is a significant increase in the visuospatial/executive scales, identification, attention, language and orientation, while there are no observable differences between abstraction and deferred memory recall. These results seem to indicate that the increased functions are mainly related to the type of training provided, which is in accordance with previous research and meta-analysis [8], with abstraction and delayed recall being the least accessed functions with this training program, thus proving that illiterate elderly people achieve similar improvements in their cognitive status to those achieved by elderly people with higher educational levels [7,8] when they undertake a cognitive training program. This study is interesting given the characteristics of the participating population, from which their ability to improve their cognitive status could be questioned. However, this work shows once again that there is cognitive plasticity in the elderly with a low educational level, as well as in elderly people with middle or higher studies [14,22]. Despite this, it should be noted that the weakness of this study lies in the low number of participants and in the bias that may exist when choosing a sample for convenience. It is difficult to find a large number of illiterate older adults gathered in a single institution, since the majority of this population group usually live in rural areas of the country and are not used to participating in mental health workshops and in a context where all the participants are women. The data obtained are considered to be a contribution to the scientific community because, despite working with a population that, according to the reviewed literature, is very likely to have brains and cognitive systems that function in an atypical way with a greater tendency to develop alterations, in brain networks and at the cognitive level [12], it was possible to intervene in cognitive functions. This can be explained in terms of cognitive plasticity, considering it as an important modulating variable in the cognitive improvement they achieved in the application of the program [7]. As was the case in other studies, the importance of cognitive plasticity in successful aging is again evident [5] as demonstrated by another meta-analysis [8], in which the importance of cognitive plasticity for memory training—also reinforced in this work—was shown. Additionally, our results show that it is possible to improve cognitive abilities, such as language, attention, visuospatial and executive abilities, and orientation. This can increase independence in daily-life activities, as our authors analyzed [23]. As a result, this study agrees with previous studies [3,8,23,24], reaffirming that older adults with MCI who participated in these studies were able to improve significantly in general cognition and in specific functions, with language being one of the abilities whose training obtained better results as previously evidenced in another work [3]. The cognitive training program Mente Sana [Healthy Mind] shows that it is effective in the treatment group for this study. The results are very positive, since they show that illiterate old adults with cognitive impairment have the ability to improve in those aspects in which they are systematically trained. This research aims to standardize it in other vulnerable populations, working in a specialized way with groups in which there is little interest, such as illiterate older adults. It is necessary to promote constant work with this population, and also with older adults who have been diagnosed with moderate or severe cognitive impairment, to continue advancing research related to the rehabilitation of cognitive functions. It is necessary to reproduce this work with other groups of illiterate older adults, including male participants, to verify the possibility of preventing cognitive decline in this population group through conducting this type of intervention, and to establish if there are different effects according to the gender of the participants.

## Figures and Tables

**Figure 1 geriatrics-05-00034-f001:**
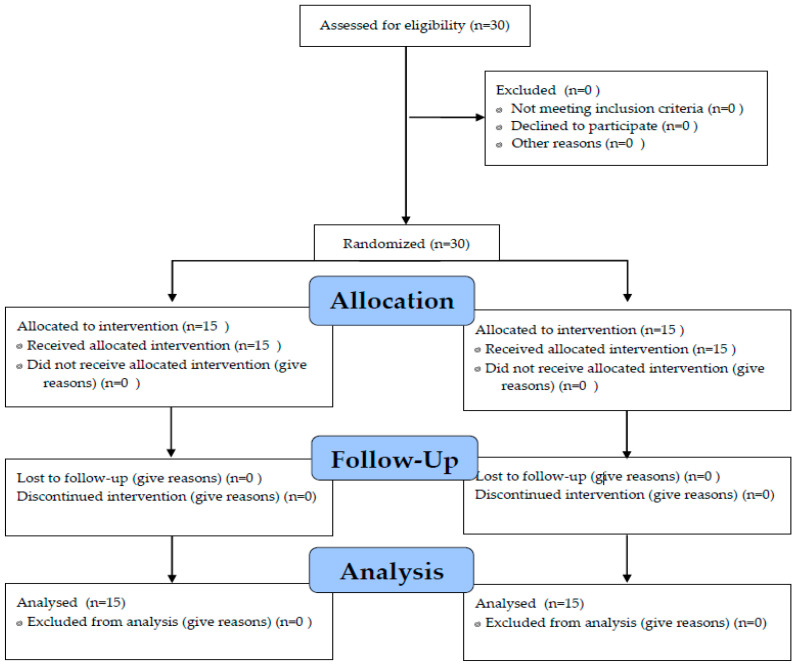
CONSORT Flow Diagram.

**Figure 2 geriatrics-05-00034-f002:**
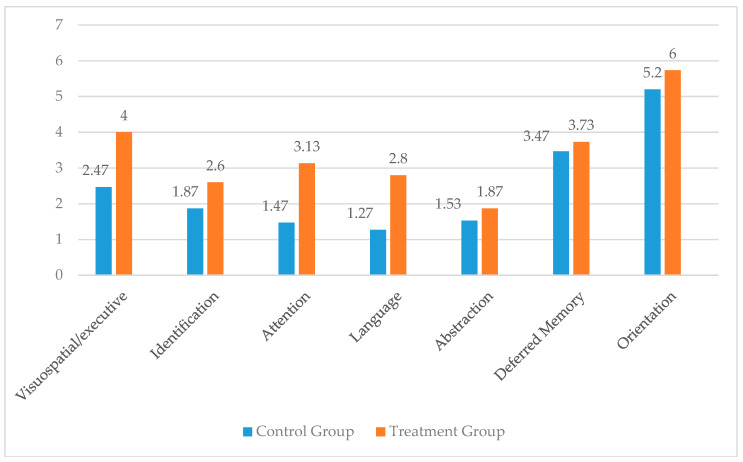
Effect of training in the different areas evaluated by MoCA in the treatment group compared to the control group.

**Table 1 geriatrics-05-00034-t001:** General Linear Model of repeated-measures. *Moment (pretest/posttest) Group (treatment/control).*

		Control G.	Treatment G		
		*n* = 15	*n* = 15	INTRAGROUP DIF.	INTERGROUPDIF.
PRETESTMOCA				GROUP	
	M	18.60	17.26	F = 14.641 **	F = 20.52 **
	S.D.	1.24	1.66	ETA = 0.343	ETA = 0.989
	RANGE	17–21	17–21	*p* = 0.958	*p* = 1.000
POSTESTMOCA				INTERACTION MOMENT x GROUP	
	M	19.00	23.87	F = 45.08 **	
	S.D.	1.33	4.32	ETA = 0.617	
	RANGE	15–20	19–29	*p* = 1.000	

** The difference is significant at 0.01 (*p* < 0.01).

**Table 2 geriatrics-05-00034-t002:** Effect of training in the different areas evaluated by MoCA in the treatment group compared to the control group.

	Group	Mean	Standard Deviation	*t*	*p*
**Visuo-Spatial/Executive**	Control	2.4667	0.63994	4.56	0.001
	Treatment	4.0000	1.13389		
**Identification**	Control	1.8667	0.74322	3.15	0.004
	Treatment	2.6000	0.50709		
**Attention**	Control	1.4667	0.51640		
	Treatment	3.1333	1.55226	3.94	0.001
**Language**	Control	1.2667	0.59362		
	Treatment	2.8000	0.41404	8.20	0.0001
**Abstraction**	Control	1.5333	0.63994		
	Treatment	1.8667	0.35187		
**Deferred Memory Recall**	Control	3.4667	0.63994		
	Treatment	3.7333	1.09978		
**Orientation**	Control	5.2000	0.67612		
	Treatment	5.7333	0.45774	2.53	0.018

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
