# Peer review of "Effectiveness of the “Mente Sana [Healthy Mind]” Cognitive Training Program for Older Illiterate Adults with Mild Cognitive Impairment"

_geriatrics, 2020, doi:10.3390/geriatrics5020034_

Round 1
Reviewer 1 Report
Thank you for the opportunity to read your paper. Cognitive training is a major issue in geriatrics. This paper reports the result of quasi-experimental research with pre- and post-test design. I have read the paper with a great interest.
Unfortunately, authors did not explain clearly enough about their study procedures or instruments.
I have pointed out some issues to be dealt with by authors before considered for publication.
Major issues:
- Did authors acquired the approval from the ethical review board?
- Please reconsider the writings in line 132-135 in page 4. It seems that authors attribute being illiterate to mere personal responsibility. However, low socio-economic status such as poverty or lack of educational opportunities are often pointed out as reasons for being illiterate and having lower cognitive functions. For me, this part is a little bit insensitive for illiterate persons.
- Please explain what is “Peruvian Social Security Health Institution”(line 155, p5) so that international readers can understand the characteristics of the study population more concretely.
- Please explain how authors adjusted each MoCA scale to the degree of instruction of the population? (line179-180, p5) . This is very important.
- Figure 1 is difficult to understand. Figures showing pre-and post-test differences by treatment status (treatment versus control) might be more informative.
- How authors calculated values in “Figure 1”? Please explain. Usually, MoCA sub scale score is in the range of 0-6, not 24.667 or 40.000.
- Authors did not present enough explanations about the program ”Mente Sana”. Please explain what kind of experts (line186, p5) validated the program, in which way? How exercises are done? In groups or individually?
- Without a detailed explanation about the program, readers cannot be convinced about its effectiveness on cognitive functions.
- Please show baseline characteristics of two groups (line200-, p5). How about gender, age, MoCA scores, and health status in each group? In pre and post test design, such information is a “must”.
- I don’t understand the Table 1. Please re-make it. Put both pre- and post-mean values, minimum and maximum values for both control and treatment groups. Also, I recommend making another table or figure from line286-291, p8-9.
- Line256-261, p7 might be a logical leap. How authors can conclude that the program was effective? Please match the writings in results section in the text and the table and the figure.
- At least, authors should write possible biases or limitations in the study.
Minor issues:
- Some wordings are difficult to understand. For example, what is “permanent assistance” in line 161, p5? Do authors mean “institutionalized”?
- In Table 1, “Med.” mean median? “Person-to-person effects” in line254, p7 is “intra group difference”? Please use terms consistently.
Author Response
Thank you for having reviewed our manuscript and for having made a sincere criticism that contributes to the proper conduct of our work to contribute to the scientific community. Regarding the translation of the text, it was prepared by a certified center and made the recommended corrections regarding the language. The observations have been corrected accordingly:
POINT 1: Did authors acquired the approval from the ethical review board?
RESPONSE 1: In lines 237-239 (p) it is explained that the investigation obtained the approval of the Ethics Committee of the Social Security of Health of Peru (EsSalud) who cares about the well-being of the Research participants made in this institution.
POINT 2: Please reconsider the writings in line 132-135 in page 4. It seems that authors attribute being illiterate to mere personal responsibility. However, low socio-economic status such as poverty or lack of educational opportunities are often pointed out as reasons for being illiterate and having lower cognitive functions. For me, this part is a little bit insensitive for illiterate persons.
Response 2: The paragraph has been corrected on lines 127-129 (p4), preserving the author's main idea and taking into account the possible sociodemographic reasons that lead to illiteracy.
POINT 3: Please explain what is “Peruvian Social Security Health Institution”(line 155, p5) so that international readers can understand the characteristics of the study population more concretely.
Response 3: The characteristics of the Peruvian Social Security Health (EsSalud) have been described on lines 150-158 (p5).
POINT 4: Please explain how authors adjusted each MoCA scale to the degree of instruction of the population? (line179-180, p5) . This is very important.
Response 4: It was explained in lines 181-186(p5) that the MoCa test adjusts its scores according to the degree of schooling in Latin America, in this case the illiterate.
POINT 5: Figure 1 is difficult to understand. Figures showing pre-and post-test differences by treatment status (treatment versus control) might be more informative.
Response 5: The data in figure 1 (p11) has been preserved with corrections in the real scores of each scale and we have created Table 2 where complementary data is shown.
POINT 6: How authors calculated values in “Figure 1”? Please explain. Usually, MoCA sub scale score is in the range of 0-6, not 24.667 or 40.000.
Response 6: In figure 1(p11), the means of the scores in the range corresponding to each scale have been corrected (it was a drafting error).
POINT 7: Authors did not present enough explanations about the program ”Mente Sana”. Please explain what kind of experts (line186, p5) validated the program, in which way? How exercises are done? In groups or individually?
Response 7: Lines 197-222 (p6) detail the characteristics of the MENTE SANA cognitive training program and its validation.
POINT 8: Without a detailed explanation about the program, readers cannot be convinced about its effectiveness on cognitive functions.
Response 8: Lines 197-222 (p6) detail the activities carried out in the application of the cognitive training program.
POINT 9: Please show baseline characteristics of two groups (line200-, p5). How about gender, age, MoCA scores, and health status in each group? In pre and post test design, such information is a “must”.
Response 9: Lines 160-167 (p5) have described the baseline characteristics of the participants in each group.
POINT 10: I don’t understand the Table 1. Please re-make it. Put both pre- and post-mean values, minimum and maximum values for both control and treatment groups. Also, I recommend making another table or figure from line286-291, p8-9.
Response 10: Table 1 was corrected with the required information and Table 2 was prepared with the data from lines 308-312(p9).
POINT 11: Line256-261, p7 might be a logical leap. How authors can conclude that the program was effective? Please match the writings in results section in the text and the table and the figure.
Response 11: Lines 290-294 (p8) were corrected by matching the data in the text with that in Table 1.
POINT 12: At least, authors should write possible biases or limitations in the study.
Response 12: Biases and limitations have been described on lines 355-361(p12).
Minor issues:
POINT 1:Some wordings are difficult to understand. For example, what is “permanent assistance” in line 161, p5? Do authors mean “institutionalized”?
Response 1: In lines 163 (p5) the term was corrected because the participants are not institutionalized.
POINT 2: In Table 1, “Med.” mean median? “Person-to-person effects” in line254, p7 is “intra group difference”? Please use terms consistently.
Response 2:Table 1 (p8) corrected the term.

Reviewer 2 Report
Thank you for the submission of your article. Here they are my comments.
The introduction is too long. Please tabulate the section related to the presentation of the findings of other studies as the literature review.
At the end of the introduction, please add the hypothesis of this study.
The process of recruitment of the samples should be described with details.
The data collection tools require some information on scoring, interpretation, translation and validity and reliability processes.
You should fill out the CONSORT checklist and find what details are missing in your report. Also the CONSORT diagram should be filled out and added to the methods that summarizes the protocol to your study.
How did you assigned the samples to the groups. How the data was collected and with what time distances?
The procedure in terms of the applied intrevention (congnitive therapy) to the groups should be described in detail to ensure of replicability of the study.
In the data analysis, you claim that repeated measure has been used. How many times the data have been collected that needed a repeated measure? more than twice? Have you checked the effect size? How?
A table for demographic characteristics is needed with comparison of characteristics between the groups.
Table 1 shows nothing of effect size calculations.
Figure 1 does not show between group comparisons for the domains of the findings. You must ask a statistician to check your data analysis and presentation style.
Discussion is so weak. You should compare each piece of findings with those of other studies. A thorough literature search is needed.
The limitations and suggestions for future studies should be added.
The conclusion with implications of findings are needed.
Author Response
Thank you for having reviewed our manuscript and for having made a sincere criticism that contributes to the proper conduct of our work to contribute to the scientific community. The observations have been corrected accordingly:
POINT 1: The introduction is too long. Please tabulate the section related to the presentation of the findings of other studies as the literature review.
RESPONSE 1: The BACKGROUND heading has been added to make the introduction better organized on line 71(p2).
POINT 2: At the end of the introduction, please add the hypothesis of this study.
RESPONSE 2. In lines 136-138 (p4), the study hypothesis was added.
POINT 3: The process of recruitment of the samples should be described with details.
RESPONSE 3. Lines 227-236(p7) describe in detail the sample recruitment process.
POINT 4: The data collection tools require some information on scoring, interpretation, translation and validity and reliability processes.
RESPONSE 4. On lines 176-181(p5), the required information about the data collection tool was added.
POINT 5: You should fill out the CONSORT checklist and find what details are missing in your report. Also the CONSORT diagram should be filled out and added to the methods that summarizes the protocol to your study.
RESPONSE 5. The CONSORT diagram was added in the PROCEDURES section. Due to ethical issues and control of results (contamination of effects among participants residing in the same place), research on cognitive training with people is not usually done as randomized studies but rather as quasi-experimental studies with intact groups or centers, applying a number of selection criteria. , as has been our case. Therefore, we do not see that it is pertinent to apply, in this case, the checklist or the CONSORT flow chart developed for clinical trials carried out in biomedicine, since it would appear 0 in many of the boxes on the chart. However, if the editors suggest listing it, we will.
POINT 6: How did you assigned the samples to the groups. How the data was collected and with what time distances?
RESPONSE 6. Lines 239-249(p7) detail the group assignment and data collection.
POINT 7: The procedure in terms of the applied intrevention (congnitive therapy) to the groups should be described in detail to ensure of replicability of the study.
RESPONSE 7. Lines 197-222(p6) detail the characteristics of the MENTE SANA cognitive training program and its validation.
POINT 8: In the data analysis, you claim that repeated measure has been used. How many times the data have been collected that needed a repeated measure? more than twice? Have you checked the effect size? How?
RESPONSE 8. Lines 292-294(p8) show the effect size and explain that they are two pre and post test measurements.
POINT 9: A table for demographic characteristics is needed with comparison of characteristics between the groups.
RESPONSE 9. Lines 160-167(p5) detail the sociodemographic characteristics of the groups and show that there is no difference between them.
POINT 10: Table 1 shows nothing of effect size calculations.
RESPONSE 10. The table 1(p8) added the effect size information.
POINT 11: Figure 1 does not show between group comparisons for the domains of the findings. You must ask a statistician to check your data analysis and presentation style.
RESPONSE 11. The information in the figure 1 (p11) was corrected.
POINT 12: Discussion is so weak. You should compare each piece of findings with those of other studies. A thorough literature search is needed.
RESPONSE 12. The discussion was improved with new bibliographic reviews. Check line 331(p11).
POINT 13: The limitations and suggestions for future studies should be added.
RESPONSE 13. In lines 356-361(p12), limitations and suggestions for future research were added.
POINT 14: The conclusion with implications of findings are needed.
RESPONSE 14. The conclusions were drawn up on lines 381-391(p12, p13).

Round 2
Reviewer 1 Report
Thank you for addressing my concerns in the revision.
However, I still have some concerns.
Please consider the following points.
Line 156: "residents" means that participants are living in these centers, or they are just visiting to attend the activities there? Please make it clear. Usually "residents" means "those living in the place".
Table 1: What is MLG? Do not use abbreviation. Also, I recommend to re-consider the table. Show the result consistently for pretest and posttest. For eample, what is "factor*group" in post test section?
Table 2: "Media" might be "Mean"?
Figure 1: punctuation error (e.g. 2,4667 might be 2.4667, this applies to all figures) Also, you can set the number of decimal places to two for the sake of readablity (e.g. 2,4667 can be 2.47・・).
Please clarify gender ratios for study participants. In abstract, you say bothe genders but in the text, you say all are women. Which is true?
Author Response
Thank you very much for reviewing our research again. Corrections were made accordingly:
POINT 1: Line 156: "residents" means that participants are living in these centers, or they are just visiting to attend the activities there? Please make it clear. Usually "residents" means "those living in the place".
RESPONSE: The term has been corrected on the line 158.
POINT 2: Table 1: What is MLG? Do not use abbreviation. Also, I recommend to re-consider the table. Show the result consistently for pretest and posttest. For eample, what is "factor*group" in post test section?
RESPONSE: The table has been corrected and the terms are better explained on the lines 348.
POINT 3: Table 2: "Media" might be "Mean"?
RESPONSE: The word was corrected and the word MEAN was used on the line 370.
POINT 4: Figure 1: punctuation error (e.g. 2,4667 might be 2.4667, this applies to all figures) Also, you can set the number of decimal places to two for the sake of readablity (e.g. 2,4667 can be 2.47・・).
RESPONSE: Figure 1 became Figure 2, the score was corrected, and the results were rounded to two decimal places on the line 378.
POINT 5: Please clarify gender ratios for study participants. In abstract, you say bothe genders but in the text, you say all are women. Which is true?
RESPONSE: The population is made up of women. Correction was made in the abstract on the line 20.
Reviewer 2 Report
Two concerns in the first review round have not been adressed:
1) In the Methods section, the CONSORT flow diagram must be filled out and added; even if this is a quasi-experimental study, it is needed:
http://www.consort-statement.org/consort-statement/flow-diagram
2) The discussion section needs improvment as you must compare your findings piece by piece by those of other studies;
Author Response
Thank you very much for reviewing our research again. Corrections were made accordingly:
POINT 1: In the Methods section, the CONSORT flow diagram must be filled out and added; even if this is a quasi-experimental study, it is needed:
http://www.consort-statement.org/consort-statement/flow-diagram
RESPONSE: The CONSORT diagram was designed on the line 264.
POINT 2 The discussion section needs improvment as you must compare your findings piece by piece by those of other studies;
RESPONSE: Bibliographic references were added in the subtitle DISCUSSION on the line 384.